# Integrating FSE and AHP to Identify Valuable Customer Needs by Service Quality Analysis

Tien-Hsiang Chang [1], Kuei-Ying Hsu [2], Hsin-Pin Fu [3,*], Ying-Hua Teng [4] and Yi-Jhen Li [3]

[1] Department of Intelligent Commerce, National Kaohsiung University of Science and Technology, Kaohsiung City 824, Taiwan; thchang@nkust.edu.tw
[2] Department of Marketing Management, Shu-Te University, Kaohsiung City 824, Taiwan; ingermarhsu@gmail.com
[3] Department of Marketing and Distribution Management, National Kaohsiung University of Science and Technology, Kaohsiung City 824, Taiwan; moopeni91441@gmail.com
[4] Graduate Institute of Management, National Kaohsiung University of Science and Technology, Kaohsiung City 824, Taiwan; i110123108@nkust.edu.tw
* Correspondence: hpfu@nkust.edu.tw; Tel.: +886-76-011-000 (ext. 34221)

**Abstract:** In this study, we explore the needs of different valuable customer groups for service quality and how limited resources are allocated to enhance service quality. Accordingly, we propose a hybrid multi-criteria decision-making (MCDM) tool that uses fuzzy synthetic evaluation (FSE) in combination with the analytic hierarchy process (AHP) to help companies enhance understanding of quantitative data (the weights of the factors that affect service quality) and qualitative information to identify valuable customers. Fifty-three experts and 304 consumers at convenience stores (CVS) comprise the data set. We employed the AHP to obtain index weights in the second step of FSE and conducted FSE to determine the importance of various valuable customer groups. The results demonstrate that different valuable customer groups have dissimilar perceptions and feelings about service quality. The findings indicate that customers between "20 to 29 years old" are the most valuable customer group and that most consumers do not care much about "problem solving". The analysis is distinct from extant work in that it examines the effect of receiving service quality from a consumer viewpoint, as we conducted a comprehensive analysis from both customer and expert perspectives.

**Keywords:** service quality; MCDM; FSE; AHP; valuable customer

## 1. Introduction

Market saturation, globalization, online developments, shorter product lifecycles, and ever-changing consumption patterns challenge today's retail industry. Amid fierce market competition, enterprises have been struggling to attract and retain customers [1,2]. Although the supply of tangible products has typically been robust—albeit the pandemic has created serious supply chain shortages—consumers have been conventionally attracted by intangible services. To be successful in this context, retailers must keep providing both goods and services of satisfactory quality [2]. In fact, service quality is critical to customer-oriented businesses [3] and is a key factor in the success of retailers and a major factor in consumer choice [4].

In the retail industry, Taiwan has the highest global density of convenience stores (CVS); they have adapted to the rapid growth of the retail market through unique chain management, mainly attracting customers by providing convenient services [5]. Taiwan has more than 10,000 CVS, which are distributed throughout the region [6]. Therefore, in this study, we examine the relationship between CVS customer groups and service quality in Taiwan. If CVS understand the priorities accorded to service quality indices by different valuable customer groups, they can target various customer groups. In doing so, they can

allocate their resources more effectively to improve service quality, customer satisfaction, and customer retention.

To assist CVS to realize the foregoing outcomes, scholars have undertaken empirical efforts to reveal crucial insights. However, most prior studies on retail service quality have utilized survey data and applied multiple regression analysis and structural equation modeling [7]. Such investigations, though, may be subject to the following problems: (1) the focus may be constrained, as scant work has considered the consumer's perspective, and (2) the accuracy and quality of research results may be challenged, as researchers cannot ascertain whether respondents expressed their literal views or were subject to social desirability [8]. To address the preceding issues, we invited experts to complete questionnaires regarding service quality through multi-criteria decision-making (MCDM) approaches [9]. Because surveys of experts are used in MCDM research methods, obtaining a consistent view is the main obstacle. In addition, identifying valuable customer groups entails using two paradigms: one is the recency, frequency, and monetary (RFM) method, and the other is through gap analysis (GA) to analyze consumers' attitudes or feelings (Ho et al., 2012) [10]. Although these methods are used to segment customers, they have the following disadvantages: (1) the RFM method requires customers to have a time period of consumption information [11], and (2) customer satisfaction is determined using GA, which can represent a valuable customer, but this relation is weak [12].

The above discussion suggests that studies utilizing feedback from customers or experts have advantages as well as disadvantages. Chen et al. [13] averred that issues surrounding evaluation should be determined using multiple perspectives, and that evaluating and commenting on each perspective individually is necessary. How to consider all perspectives and conduct a comprehensive review, though, is a synthetic evaluation issue. Therefore, Chen et al. [13] proposed fuzzy synthetic evaluation (FSE) to resolve the synthetic evaluation issue of multiple perspectives.

FSE is a widely used method in fuzzy mathematics; the MCDM approach is based on fuzzy mathematics and centered on the grade of the membership function of fuzzy mathematics [14,15]. FSE transfers the qualitative evaluation into the quantitative evaluation to offer a synthetic evaluation and then implements an evaluation of a single index. It then conducts a comprehensive fuzzy evaluation of all indices to prevent omission of any statistical information and information loss [16]. This approach addresses deviation from objective truth and identifies certainty evaluations with "Yes" or "No".

Using FSE to segment customers does not require a customer's time period of behavior. Rather, it provides the relationship between the customer's value and his/her satisfaction. In addition, implementing FSE attends to the dual opinion of customers and experts. Therefore, FSE is suitable for studies that identify valuable customer groups by service quality based on consumer and expert views.

In the second step of the FSE calculation, based on the opinions of the customers, the weights of the factors that influence service quality are then obtained by using the analytic hierarchy process (AHP) [17]. In this study, we investigated the needs of different valuable customer groups for service quality in efforts to improve customer satisfaction. We propose a hybrid MCDM tool, which uses FSE in combination with AHP, to understand the quantitative data and the qualitative information to identify valuable customers. That is, this study integrates the MCDM method (FSE and AHP) based on consumer and expert opinions as an enhanced way to determine the difference from only a customer perspective based on traditional regression methods.

Toward this end, to identify CVS's valuable customer groups that have different levels of service quality, we first conducted an objective evaluation based on expert views of CVS and then judged service quality based on consumers' experience. Generally, store owners or academic studies have argued that youth are CVS's main customers; however, CVS do not have a membership system and so cannot easily understand the needs of different age groups. To engage in market segmentation further, this paper, therefore, divided the youth into different age groups—not only to identify accurately the valuable customer

groups, but also to understand the levels of service quality for various age groups. This undertaking should help CVS develop marketing plans with enhanced precision, thus affording the use of their limited resources efficaciously.

## 2. Literature Review

### 2.1. Service Quality Index

SERVQUAL [18] primarily applies to financial services and services that require frequent interactions with customers. Though many scholars have used SERVQUAL in the retail sector, the results have failed to demonstrate the features of the retail sector. Unlike financial services, the retail sector includes the sale of tangible products. Along with services, this represents the major business activity of retail [19]. Dabholkar et al. [19] developed a dedicated scale—namely, the retail service quality scale (RSQS); it provides a better measure of retail service quality than SERVQUAL. Unlike SERVQUAL, RSQS has a hierarchical structure and sub-aspects in which physical elements include appearance and convenience. "Reliability" represents a "promise of doing it right", and "personal interaction" encompasses "inspiring confidence" and "being courteous/helpful".

Subsequently, researchers began applying RSQS to assess the impact of retail service quality on customer value [2,20,21]. For example, Martinelli and Balboni [20] utilized RSQS to explore the relationship between the loyalty of 450 grocery customers and service quality. Sivapalan and Jebarajakirthy [2] employed RSQS to analyze the relation between the loyalty of 2375 supermarket customers and service quality in Sri Lanka. Moreover, scholars surveyed 600 retail outlet customers in Delhi, India, using RSQS to examine the relationship between service quality and customer satisfaction [21].

The foregoing investigations collected information from consumers and applied quantitative research methods. That work revealed that retail service quality could be effectively matched with customers' needs, as using RSQS helps identify valuable customers and their needs. In addition, some scholars have utilized people, process, and physical evidence [22] to investigate the service quality of different industries [23], thus indicating that these three Ps can be used to examine service quality.

### 2.2. Identifying the Valuable Customer Group of CVS

Gil-Saura and Ruiz-Molina [24] revealed the existence of two different profiles of customers in retail commerce. They did so by characterizing these groups of retail customers based on their perceptions of the benefits and costs derived from their relations with retailers. To segment retail customers, Wong and Sohal [25]—using 1261 surveys administered to shoppers leaving a large chain departmental store in Australia—found that service quality was positively associated with customer loyalty in retail relationships.

As noted above, in the extant literature, two methods have generally been used to segment retail customers: one is RFM, and the other is GA. Chan [26] applied these methods to identify valuable customers for an automobile retailer in Taiwan. Using GA, Tsai et al. [27] identified potential service quality gaps to improve customer satisfaction. Safari et al. [28] utilized the RFM model in an IT company in Iran to segment customers.

Despite their use in prior work, both RFM and GA segmentation methods still have disadvantages. The RFM method requires customers to have a time period of consumption information. GA cannot provide the strength of the relationship between customer satisfaction and customer value. Using FSE to segment customers does not require a time period for customers' data, and it does provide the degree of the relationship between customers and experts of retailing.

### 2.3. FSE Method

MCDM is a feasible approach for generating the multi-dimensional and hierarchical aspects of service quality [19]. For example, Chiu et al. [29], based on weights formed from experts' views, applied MCDM to analyze expert views on the service quality of electronic stores. Using consumers' views, Gopalan and Satpathy [9] employed fuzzy AHP

to evaluate the service quality of retailers in India and ranked three retailers according to service quality. Kuo [30] proposed a new interval-valued fuzzy MCDM method to assess the service quality of China Channel Airways. He did so to help the company identify the strengths and weaknesses of specific passenger services, attend to the weaknesses, and further improve its service quality. Perçin [31] combined three MCDM methods to assess the service quality of Turkish Airlines.

The MCDM method has also been widely applied to such areas as examining the relation between perceived service quality and profitability [32], analyzing restaurant service quality [33], and investigating power distribution service quality [34]. Moreover, abundant research has proposed a hybrid MCDM method for use in a variety of other fields—such as risk analysis methods in water distribution networks [35], assessment of distance-learning modules [36], resource selection in mobile crowd computing [37], and the ranking of evaluations of road sections [38].

To address the aforementioned deficiencies of the traditional method for determining valuable customers, some scholars have used FSE in decision making in various contexts, including enhancement of customer value through amelioration of store image [16], customer satisfaction [39], and optimal facility management strategy for commercial office buildings [40]. In the present study, we applied FSE to collect and combine the views of consumers and experts to enhance understanding of the level of service quality awareness among different customer groups, assist CVS to overcome their service weaknesses, facilitate CVS in allocating limited resources for major indices in an appropriate manner, and improve their service quality and competitiveness.

## 3. Methodology

Many problems are fuzzy and cannot be quantified, such as consumer satisfaction and loyalty, as well as consumers' feelings regarding service quality (the focus of this study). However, based on individual criteria, the human brain automatically classifies complex problems according to experience or logic. Toward this end, one or more mathematical methods can be applied to address this issue with enhanced objectivity. When a problem involves several indices, the judging criteria and measured values can be fuzzified so that FSE can precisely assess the problem. That is, FSE transfers qualitative evaluation into quantitative evaluation through fuzzy mathematics. Thus, to conduct an overall evaluation, synthetic and general evaluations must be implemented for the relative indices. In this study, we applied FSE to discuss CVS service quality-related topics according to the following steps: develop the index set, weight set, and evaluation set; construct the fuzzy evaluation matrix; and implement FSE and defuzzification (described below):

(1) Develop the index set: The index set contains various indices that affect the evaluation target. Equation (1):

$$U_i = \{u_1,\ u_2, \ldots, u_n\},\ i = 1,\ 2, \ldots,\ n, \tag{1}$$

where $U$ represents the index set and $u_i$ represents each index. Generally, these indices have different degrees of fuzziness.

(2) Develop the weight set: Because the indices in the index set have different levels of significance, to reflect the level of significance of each index, each index $u_i$ is assigned a weight $W_j$. Equation (2):

$$W_i = \{w_1,\ w_2,\ \ldots, w_n\},\ i = 1,\ 2, \ldots,\ n. \tag{2}$$

The weight set used in this study was developed based on consumers' AHP questionnaires. For a detailed discussion of the AHP (Appendix A), please refer to [17].

(3) Develop the evaluation set: The evaluation set contains the evaluation results that an evaluate (i.e., evaluator) may make of the evaluation target. Equation (3):

$$V_j = \{v_1,\ v_2, \ldots, v_m\},\ j = 1,\ 2,\ \ldots,\ m, \tag{3}$$

where $v_j$ represents the possible evaluation result.

(4) Construct the fuzzy evaluation matrix: $r_{ij}$ is a single index evaluation set; that is, it is the evaluation made by evaluate $j$ on index $i$ to determine the grade of membership of the evaluation target in the evaluation set. Therefore, it can be known that a single index evaluation set is a fuzzy subset in the evaluation set, and single index evaluation sets R can be integrated into the fuzzy evaluation matrix, as depicted in Equation (4).

$$R = \left(r_{ij}\right)_{n\times m} = \begin{bmatrix} r_{11} & r_{12} & \cdots & r_{1m} \\ r_{21} & r_{22} & \cdots & r_{2m} \\ \vdots & \vdots & \ddots & \vdots \\ r_{n1} & r_{n2} & \cdots & r_{nm} \end{bmatrix}; \; i = 1,\,2,\,\ldots,\,n;\, j = 1,\,2,\,\ldots,\,m \quad (4)$$

(5) Implement FSE: Because a single index fuzzy evaluation only represents the evaluation of a single index and cannot provide the evaluation of all indices, in this study we aimed to understand the synthetic effects of indices. Although the sum of single index evaluation results can reflect the synthetic effects of all indices, we cannot know the level of significance of each index. Thus, we considered the weight set. Shown in Equation (5) is the FSE.

$$_c D_j = W{\cdot}R = [w_1,\, w_2,\, \ldots w_n]{\cdot}\begin{bmatrix} r_{11} & r_{12} & \cdots & r_{1m} \\ r_{21} & r_{22} & \cdots & r_{2m} \\ \vdots & \vdots & \ddots & \vdots \\ r_{n1} & r_{n2} & \cdots & r_{nm} \end{bmatrix} = (d_1,\, d_2,\ldots,d_m); \quad (5)$$
$$i = 1,\,2,\,\ldots,\,n;\, j = 1,\,2,\,\ldots,\,m$$

$D_j$ is the FSE set, and $d_1, d_2, \ldots,$ and $d_m$ are the FSE indices (also called decision-making set). It also represents the membership grade of the evaluation target to index j in the evaluation set under the synthetic effects of all indices.

(6) Defuzzification: Defuzzification transfers the FSE set $D_j$ into explicit values. We used weighted means to standardize the FSE indices. The formula is as follows:

$$K = D \cdot S$$

where K represents the defuzzification score, which can be obtained by multiplying the FSE set D by the evaluation score set S. S= [100 80 60 40 20] and is the final evaluation score of an index.

The second step of FSE is applied to develop the weight set. Generally, the AHP, as proposed by Saaty [17], is widely utilized to obtain index weights [16]. The weights present analysis results according to "weight" and "ranking", but they cannot demonstrate the correlation among or significance of the multiple views of groups. Toward this end, in this study, we developed the weight set based on the combination of expert views and AHP.

## 4. Application of FSE

In this study, we focused on finding different valuable customer groups based on service quality. Per the literature review, RSQS questionnaires [19] and service evidence of the three Ps (people, process, and physical evidence) [22] can form the theoretical framework of service quality. Therefore, five scholars familiar with service quality were invited to discuss and develop the retail service quality hierarchy of this study according to the correlation between service quality indices based on the RSQS questionnaires and service evidence of the three Ps. Finally, three policy-related questions were then included into four other aspects of the questionnaire, as shown in Table 1.

**Table 1.** CVS service quality three-layer hierarchical indices.

| Criteria | Indices |
|---|---|
| Physical Aspects (S1) | Modern-looking equipment and fixtures (S11)<br>Physical facilities are visually appealing (S12)<br>Materials associated with store service are visually appealing (S13)<br>Clean, attractive, and convenient public areas (S14) |
| Reliability (S2) | This store provides its services at the time it promises to do so (S21)<br>This store insists on error-free sales transactions and records (S22)<br>This store has merchandise available when the customers want it (S23)<br>When this store promises to do something by a certain time, it will do so (S24) |
| Personal Interaction (S3) | Employees in this store have the knowledge to answer customers' questions (S31)<br>Employees in this store give prompt service to customers (S32)<br>Employees in this store are consistently courteous with customers (S33)<br>Employees in this store tell customers exactly when services will be performed (S34) |
| Problem Solving (S4) | Willingness to handle returns and exchanges (S41)<br>Sincere interest to solve problems (S42)<br>Handling complaints directly and immediately (S43) |

Note: Goal criteria, sub-criteria, and indices have a unique code, as the code of "Physical Aspects" is S1.

In this study, we employed a questionnaire according to the two-stage appraisal model. In Stage 1, retail experts with more than five years of experience in CVS service quality operations across Taiwan were asked to complete the questionnaire. The survey results were used as the "evaluation set" of FSE. Thus, experts' input could be utilized for the overall evaluation of the indices.

In Stage 2, CVS consumers across Taiwan participated in the study. Consumers were selected using convenience sampling of social media and then grouped into "under 19 years old", "20 to 29 years old", "30 to 39 years old", and "over 40 years old". The purposes of this stage were to (1) use the consistent results obtained through the AHP questionnaire analysis as the weight set of FSE, and (2) analyze the differences in the considerations of different customer groups regarding CVS service quality.

Below is the 7-step FSE process employed in this study.

(1) Develop the index set: Main aspect index set U = {S1, S2, S3, S4}; Sub-aspect index set U S1 = {S11, S12, S13, S14}; U S2 = {S21, S22, S23, S24}; U S3 = {S31, S32, S33, S34}; U S4 = {S41, S42, S43}. Note: The content of these codes is presented in Table 1. "S1" denotes physical aspects.

(2) Develop the weight set: A total of 304 valid consumer questionnaires and 53 expert questionnaires were obtained for analysis and manual screening. We implemented the AHP analysis of valid consumer questionnaires based on the retail service quality indices and derived the weights for the CVS service quality indices, as depicted in Table 2. Shown in Table 3 are the weights and ranking of CVS service quality indices by the four customer groups. Delbecq et al. [41] claimed that 15 to 30 individuals is a reasonable sample size if the group of experts is highly homogeneous. Therefore, the preliminary image can be adequately plotted using the opinions of the 53 experts, as our expert respondents were senior managers in retailing. Therefore, our sample size of 53 experts was suitable for this study.

**Table 2.** Weights of CVS service quality three-layer indices.

| Criteria Weights (A) | Indices | Local Weights (B) | Global Weights (C) (C = A × B) |
|---|---|---|---|
| S1 0.3117 | S11 | 0.2844 | 0.0886 |
| | S12 | 0.2054 | 0.0640 |
| | S13 | 0.2819 | 0.0879 |
| | S14 | 0.2283 | 0.0712 |
| S2 0.2961 | S21 | 0.3427 | 0.1015 |
| | S22 | 0.2844 | 0.0842 |
| | S23 | 0.1950 | 0.0577 |
| | S24 | 0.1778 | 0.0526 |
| S3 0.2099 | S31 | 0.3583 | 0.0752 |
| | S32 | 0.2677 | 0.0562 |
| | S33 | 0.2020 | 0.0424 |
| | S34 | 0.1720 | 0.0361 |
| S4 0.1822 | S41 | 0.3741 | 0.0682 |
| | S42 | 0.3817 | 0.0695 |
| | S43 | 0.2442 | 0.0445 |

Note 1: A: Initial weights of criteria level; B: initial weight of sub-criteria level; C: normalized weights of criteria (C = A × B). Note 2: All C.R. values are less than 0.1; this indicates that the entire hierarchy has good consistency of every layer.

**Table 3.** Weights of CVS service quality indices by customer group.

| Under 19 Years Old | | 20 to 29 Years Old | | 30 to 39 Years Old | | Over 40 Years Old | |
|---|---|---|---|---|---|---|---|
| Weights | Sub-Criteria | Weights | Sub-Criteria | Weights | Sub-Criteria | Weights | Sub-Criteria |
| 0.3688 | S1 | 0.3040 | S2 | 0.2825 | S1 | 0.3305 | S1 |
| 0.2859 | S2 | 0.2927 | S1 | 0.2734 | S2 | 0.3018 | S2 |
| 0.2019 | S3 | 0.2036 | S3 | 0.2658 | S3 | 0.2228 | S3 |
| 0.1435 | S4 | 0.1996 | S4 | 0.1783 | S4 | 0.1449 | S4 |
| Weights | Index | Weights | Index | Weights | Index | Weights | Index |
| 0.1350 | S11 | 0.1003 | S21 | 0.1113 | S21 | 0.1136 | S21 |
| 0.0950 | S21 | 0.0867 | S22 | 0.0965 | S31 | 0.1001 | S11 |
| 0.0853 | S12 | 0.0859 | S13 | 0.0859 | S13 | 0.0897 | S31 |
| 0.0842 | S13 | 0.0775 | S11 | 0.0820 | S11 | 0.0893 | S13 |
| 0.0799 | S22 | 0.0747 | S42 | 0.0807 | S22 | 0.0888 | S22 |
| 0.0715 | S31 | 0.0733 | S41 | 0.0750 | S32 | 0.0728 | S12 |
| 0.0692 | S42 | 0.0720 | S14 | 0.0735 | S42 | 0.0682 | S14 |
| 0.0642 | S14 | 0.0706 | S31 | 0.0622 | S41 | 0.0590 | S32 |
| 0.0600 | S23 | 0.0607 | S23 | 0.0622 | S14 | 0.0585 | S41 |
| 0.0537 | S32 | 0.0573 | S12 | 0.0525 | S12 | 0.0567 | S42 |
| 0.0510 | S24 | 0.0563 | S24 | 0.0493 | S33 | 0.0522 | S23 |
| 0.0428 | S33 | 0.0562 | S32 | 0.0450 | S34 | 0.0472 | S24 |
| 0.0415 | S41 | 0.0515 | S43 | 0.0445 | S24 | 0.0408 | S33 |
| 0.0340 | S34 | 0.0412 | S33 | 0.0426 | S43 | 0.0334 | S34 |
| 0.0328 | S43 | 0.0355 | S34 | 0.0369 | S33 | 0.0297 | S43 |

The weight set used in this study was developed based on customer AHP questionnaires. The weights obtained by the AHP (rounded off to four decimal places) were used to establish the weight values (for the AHP calculation process see Saaty [17] for detail). Weights by customer groups are shown in Table 6.

(3) Develop the evaluation set: Evaluation set V = {Strongly agree, Agree, Neutral, Disagree, Strongly disagree}.

(4) Construct the fuzzy evaluation matrix: The views of the 53 experts on CVS service quality are listed in Table 4 by scores that are standardized and transferred to the fuzzy

evaluation matrix, as shown from $R_1$ to $R_4$. The fuzzy evaluation matrix transferred from Table 4 is depicted in Equations (6)–(9).

$$R_1 = \begin{bmatrix} 0.4340 & 0.4528 & 0.1132 & 0.0000 & 0.0000 \\ 0.4528 & 0.4340 & 0.0943 & 0.0189 & 0.0000 \\ 0.6226 & 0.3019 & 0.0566 & 0.0000 & 0.0189 \\ 0.6226 & 0.3019 & 0.0755 & 0.0000 & 0.0000 \end{bmatrix} \tag{6}$$

$$R_2 = \begin{bmatrix} 0.7547 & 0.2453 & 0.0000 & 0.0000 & 0.0000 \\ 0.5849 & 0.3774 & 0.0189 & 0.0000 & 0.0189 \\ 0.6038 & 0.3208 & 0.0755 & 0.0000 & 0.0000 \\ 0.6226 & 0.3396 & 0.0377 & 0.0000 & 0.0000 \end{bmatrix} \tag{7}$$

$$R_3 = \begin{bmatrix} 0.6792 & 0.2830 & 0.0377 & 0.0000 & 0.0000 \\ 0.6604 & 0.2642 & 0.0566 & 0.0000 & 0.0189 \\ 0.6038 & 0.3208 & 0.0566 & 0.0000 & 0.0189 \\ 0.5849 & 0.3208 & 0.0943 & 0.0000 & 0.0000 \end{bmatrix} \tag{8}$$

$$R_4 = \begin{bmatrix} 0.3585 & 0.4340 & 0.1887 & 0.0000 & 0.0189 \\ 0.4906 & 0.3208 & 0.1887 & 0.0000 & 0.0000 \\ 0.5094 & 0.3962 & 0.0755 & 0.0000 & 0.0189 \end{bmatrix} \tag{9}$$

(5)  Implement FSE: The FSE set D (the decision-making set) was obtained for each of the four customer groups by multiplying $W_{a1}$ to $W_{a4}$ by $R_1$ to $R_4$ ($W_{bi}$, $W_{ci}$, and $W_{di}$ follow the same procedure), respectively, and is expressed in $D_a$, $D_b$, $D_c$, and $D_d$, as illustrated in Equation (10).

$$
\begin{aligned}
D_{a1} &= W_{a1} \times R_1 \\
&= (2314, 0.2284, 0.1741) \times \begin{bmatrix} 0.4340 & 0.4528 & 0.1132 & 0.0000 & 0.0000 \\ 0.4528 & 0.4340 & 0.0943 & 0.0189 & 0.0000 \\ 0.6226 & 0.3019 & 0.0566 & 0.0000 & 0.0189 \\ 0.6226 & 0.3019 & 0.0755 & 0.0000 & 0.0000 \end{bmatrix} \\
&= [0.5143 \ 0.3877 \ 0.0893 \ 0.0044 \ 0.0043]
\end{aligned} \tag{10}
$$

The FSE index (the decision-making set) $D_a$ was obtained by processing $D_{a2}$, $D_{a3}$, and $D_{a4}$ following the same procedure as $D_{a1}$, as shown in Equation (11).

$$\begin{bmatrix} 0.5143 & 0.3877 & 0.0893 & 0.00440 & 0.0043 & D_{a1} \\ 0.6520 & 0.3149 & 0.0278 & 0.0000 & 0.0053 & D_{a2} \\ 0.6424 & 0.2923 & 0.0563 & 0.0000 & 0.0090 & D_{a3} \\ 0.4567 & 0.3707 & 0.1628 & 0.0000 & 0.0098 & D_{a4} \end{bmatrix} \tag{11}$$

$D_b$, $D_c$, and $D_d$ for the other three customer groups were obtained by following the same procedure, as illustrated in Equations (12)–(14).

$$\begin{bmatrix} 0.5394 & 0.3677 & 0.0836 & 0.0037 & 0.0055 & D_{b1} \\ 0.6517 & 0.3155 & 0.0274 & 0.0000 & 0.0054 & D_{b2} \\ 0.6423 & 0.2920 & 0.0566 & 0.0000 & 0.0090 & D_{b3} \\ 0.4469 & 0.3818 & 0.1594 & 0.0000 & 0.0118 & D_{b4} \end{bmatrix} \tag{12}$$

$$\begin{bmatrix} 0.5364 & 0.3702 & 0.0842 & 0.0035 & 0.0057 & D_{c1} \\ 0.6627 & 0.3098 & 0.0219 & 0.0000 & 0.0056 & D_{c2} \\ 0.6439 & 0.2639 & 0.0773 & 0.0006 & 0.0067 & D_{c3} \\ 0.4490 & 0.3818 & 0.1594 & 0.0000 & 0.0118 & D_{c4} \end{bmatrix} \tag{13}$$

$$\begin{bmatrix} 0.5281 & 0.3767 & 0.0860 & 0.0042 & 0.0051 & D_{d1} \\ 0.6580 & 0.3120 & 0.0245 & 0.0000 & 0.0056 & D_{d2} \\ 0.6463 & 0.2906 & 0.0547 & 0.0000 & 0.0084 & D_{d3} \\ 0.4410 & 0.3819 & 0.1655 & 0.0000 & 0.0115 & D_{d4} \end{bmatrix} \tag{14}$$

(6)  Defuzzification: Defuzzification helps obtain the target score and overall evaluation score of each customer group. It entails two steps.

(i)  The target defuzzification score of each customer group: The FSE index ($D$), as obtained in the previous step, was defuzzified. That is, FSE indices $D_a$, $D_b$, $D_c$, and $D_d$ were multiplied by the evaluation score set S = $\begin{bmatrix} 100 & 80 & 60 & 40 & 20 \end{bmatrix}$ [16] to obtain the target score of each of the four customer groups. The formula is presented in Equation (15).

$$\begin{bmatrix} 0.5143 & 0.3877 & 0.0893 & 0.0440 & 0.0043 & D_{a1} \\ 0.6520 & 0.3149 & 0.0278 & 0.0000 & 0.0053 & D_{a2} \\ 0.6424 & 0.2923 & 0.0563 & 0.0000 & 0.0090 & D_{a3} \\ 0.4567 & 0.3707 & 0.1628 & 0.0000 & 0.0098 & D_{a4} \end{bmatrix} \times \begin{bmatrix} 100 \\ 80 \\ 60 \\ 40 \\ 20 \end{bmatrix} = \begin{bmatrix} 88.07 \\ 92.17 \\ 91.18 \\ 85.29 \end{bmatrix} \tag{15}$$

FSE scores for the other three customer groups were obtained similarly, as shown in Equations (16)–(18).

$$\begin{bmatrix} 0.5394 & 0.3677 & 0.0836 & 0.0037 & 0.0055 & D_{b1} \\ 0.6517 & 0.3155 & 0.0274 & 0.0000 & 0.0054 & D_{b2} \\ 0.6423 & 0.2920 & 0.0566 & 0.0000 & 0.0090 & D_{b3} \\ 0.4469 & 0.3818 & 0.1594 & 0.0000 & 0.0118 & D_{b4} \end{bmatrix} \times \begin{bmatrix} 100 \\ 80 \\ 60 \\ 40 \\ 20 \end{bmatrix} = \begin{bmatrix} 88.64 \\ 92.16 \\ 91.17 \\ 85.03 \end{bmatrix} \tag{16}$$

$$\begin{bmatrix} 0.5364 & 0.3702 & 0.0842 & 0.0035 & 0.0057 & D_{c1} \\ 0.6627 & 0.3098 & 0.0219 & 0.0000 & 0.0056 & D_{c2} \\ 0.6439 & 0.2639 & 0.0773 & 0.0006 & 0.0067 & D_{c3} \\ 0.4490 & 0.3818 & 0.1594 & 0.0000 & 0.0118 & D_{c4} \end{bmatrix} \times \begin{bmatrix} 100 \\ 80 \\ 60 \\ 40 \\ 20 \end{bmatrix} = \begin{bmatrix} 88.56 \\ 92.48 \\ 90.78 \\ 85.03 \end{bmatrix} \tag{17}$$

$$\begin{bmatrix} 0.5281 & 0.3767 & 0.0860 & 0.0042 & 0.0051 & D_{d1} \\ 0.6580 & 0.3120 & 0.0245 & 0.0000 & 0.0056 & D_{d2} \\ 0.6463 & 0.2906 & 0.0547 & 0.0000 & 0.0084 & D_{d3} \\ 0.4410 & 0.3819 & 0.1655 & 0.0000 & 0.0115 & D_{d4} \end{bmatrix} \times \begin{bmatrix} 100 \\ 80 \\ 60 \\ 40 \\ 20 \end{bmatrix} = \begin{bmatrix} 88.37 \\ 92.34 \\ 91.33 \\ 84.81 \end{bmatrix} \tag{18}$$

(ii)  The overall evaluation scores: After the FSE index of each of the four customer groups are defuzzified, the overall obtained score represents the goodness of fit between each customer group and the CVS service quality aspects, where greater goodness of fit reflects a more valuable customer group vis-à-vis CVS service quality. The overall FSE scores are shown in Table 5.

**Table 4.** Experts' evaluation score of CVS service quality.

| Sub-Criteria | Index | Evaluation Score | | | | | Total |
|---|---|---|---|---|---|---|---|
| | | Strongly Agree | Agree | Neutral | Disagree | Strongly Disagree | |
| S1 2.4057 * | S11 | 23 | 24 | 6 | 0 | 0 | 53 |
| | S12 | 24 | 23 | 5 | 1 | 0 | 53 |
| | S13 | 33 | 16 | 3 | 0 | 1 | 53 |
| | S14 | 33 | 16 | 4 | 0 | 0 | 53 |
| S2 2.5849 * | S21 | 40 | 13 | 0 | 0 | 0 | 53 |
| | S22 | 31 | 20 | 1 | 0 | 1 | 53 |
| | S23 | 32 | 17 | 4 | 0 | 0 | 53 |
| | S24 | 33 | 18 | 2 | 0 | 0 | 53 |
| S3 2.5236 * | S31 | 36 | 15 | 2 | 0 | 0 | 53 |
| | S32 | 35 | 14 | 3 | 0 | 1 | 53 |
| | S33 | 32 | 17 | 3 | 0 | 1 | 53 |
| | S34 | 31 | 17 | 5 | 0 | 0 | 53 |
| S4 2.2390 * | S41 | 19 | 23 | 10 | 0 | 1 | 53 |
| | S42 | 26 | 17 | 10 | 0 | 0 | 53 |
| | S43 | 27 | 21 | 4 | 0 | 1 | 53 |

* Sub-criteria index of experts: "Strongly agree" received 3 points; "Agree," 2 points; "neutral," 1 point; "disagree," −2 points; and "strongly disagree," 3 points. All indexes were the mean of points, which produced the mean of the sub-criteria index.

**Table 5.** Fuzzy synthetic evaluation scores of customer groups.

| Sub-Criteria | "Below 19 Years Old" | "20 to 29 years Old" | "30 to 39 Years Old" | "Above 40 Years Old" | Average |
|---|---|---|---|---|---|
| S1 | 88.07 | 88.64 | 88.56 | 88.37 | 88.41 |
| S2 | 92.17 | 92.16 | 92.48 | 92.34 | 92.29 |
| S3 | 91.18 | 91.17 | 90.78 | 91.33 | 91.12 |
| S4 | 85.29 | 85.03 | 85.03 | 84.81 | 85.04 |
| Mean | 89.18 | 89.25 | 89.21 | 89.21 | 89.21 |

**Table 6.** Customer groups' FSE weight sets.

| | under 19 years old | 20 to 29 years old |
|---|---|---|
| (S1–S4) | $W_a$ = (0.3688, 0.2859, 0.2019, 0.1435) | $W_b$ = (0.2927, 0.3040, 0.2036, 0.1996) |
| (S11–S14) | $W_{a1}$ = (0.3661, 0.2314, 0.2284, 0.1741) | $W_{b1}$ = (0.2647, 0.1958, 0.2935, 0.2460) |
| (S21–S24) | $W_{a2}$ = (0.3324, 0.2795, 0.2098, 0.1783) | $W_{b2}$ = (0.3298, 0.2853, 0.1996, 0.1853) |
| (S31–S34) | $W_{a3}$ = (0.3539, 0.2660, 0.2119, 0.1682) | $W_{b3}$ = (0.3470, 0.2761, 0.2025, 0.1744) |
| (S41–S43) | $W_{a4}$ = (0.2892, 0.4823, 0.2285) | $W_{b4}$ = (0.3674, 0.3743, 0.2582) |
| | **30 to 39 years old** | **over 40 years old** |
| (S1–S4) | $W_c$ = (0.2825, 0.2734, 0.2658, 0.1783) | $W_d$ = (0.3305, 0.3018, 0.2228, 0.1449) |
| (S11–S14) | $W_{c1}$ = (0.2901, 0.1858, 0.3041, 0.2200) | $W_{d1}$ = (0.3030, 0.2203, 0.2703, 0.2064) |
| (S21–S24) | $W_{c2}$ = (0.4072, 0.2952, 0.1350, 0.1626) | $W_{d2}$ = (0.3763, 0.2944, 0.1730, 0.1563) |
| (S31–S34) | $W_{c3}$ = (0.3629, 0.2822, 0.1854, 0.1694) | $W_{d3}$ = (0.4025, 0.2646, 0.1832, 0.1497) |
| (S41–S43) | $W_{c4}$ = (0.3491, 0.4120, 0.2389) | $W_{d4}$ = (0.4038, 0.3913, 0.2048) |

Note: W set represented weights set of S in Table 3.

According to the overall evaluation results (Table 5), the four valuable customer groups were ranked by the goodness of fit with CVS service quality in descending order: "20 to 29 years old" (89.25) > "30 to 39 years old" and "over 40 years old" (89.21) > "under 19 years old" (89.19). Therefore, "20 to 29 years old" customers represented the most valuable customer group: they had the highest recognition of the service quality for CVS, followed by the "30 to 39 years old" customers. The least important customers were those in the "under 19 years old" group. The four goodness of fit scores denoted the degree of customer value. Further, service quality recognition differed across the various customer groups.

## 5. Results

The results (Table 5) indicate that the goodness of fit with CVS service quality is in descending order: "20 to 29 years old" > "30 to 39 years old" and "over 40 years old" > "under 19 years old". Moreover, the important indices of the sub-criteria are as follows: "reliability" (S2) with a score of 92.29, "personal interaction" (S3) with a score of 91.12, "physical aspects" (S1) with a score of 88.41, and "problem solving" (S4) with a score of 85.04. According to Table 3, among the targeted indices, all customers perceived that the most important indices pertained to "physical aspects" (S1) with a weight of 31.17%, "reliability" (S2) with a weight of 29.61%, "personal interaction" with a weight of 20.99%, and "problem solving" with a weight of 18.22%. These findings suggest that the findings of this research using the opinions of both experts and customers differed markedly from those derived only from the opinions of customers (Table 7).

**Table 7.** Summarizing the weights of criteria on consumers and FSE.

| Sub-Criteria | Consumers | Experts | FSE (Consumers and Experts) |
| --- | --- | --- | --- |
| S1 (Physical Aspects) | 0.3117 | 2.4057 | 88.41 |
| S2 (Reliability) | 0.2961 | 2.5849 | 92.29 |
| S3 (Personal Interaction) | 0.2099 | 2.5236 | 91.12 |
| S4 (Problem Solving) | 0.1822 | 2.2390 | 85.04 |

To deeply analyze the results, the authors interviewed five CVS managers to understand possible reasons for the findings. They stated that the CVS business has unified requirements and regulations for physical aspects (physical, facility, and clothing appearance), and they constitute the most basic requirements. Moreover, the CVS managers expressed that there are more customer complaints related to issues of reliability and personal interaction than to physical aspects. Therefore, the findings of this study are consistent with the reality in Taiwanese CVS.

*Our Work Offers Four Practical Implications*

(1) Although some previous studies have provided the weights of various service quality indicators, they could not confirm whether improving the quality of each service aspect had the same effect on all consumers. An effective method (FSE) for dividing consumers into groups and then determining a CVS' most important (or valuable) customers is readily available. The denouement is market re-segmentation proposed in marketing theory.

(2) In studies of service quality and customer groups from a consumer's perspective, researchers have generally obtained consumers' views based on their set indices. Accordingly, that research scope was constrained by use of a certain theoretical framework. In this study, we applied FSE to combine consumer *and* expert views to present more exhaustive and practical results than those prior studies that were predicated only on capturing consumer perspectives.

(3) We identified the priorities accorded by different valuable customer groups to CVS service quality indices (quantitative) (Table 5). The results can provide a basis for

CVS to improve their service quality and competitiveness through the ameliorated utilization of limited resources. Such resource allocations can be predicated on the weights of different service quality indices (Table 3) in different valuable customer groups (Table 5).

(4) In this study, comparing across four service factors, we found that consumers are not especially focused on "problem solving". The possible reason is that many service procedures in CVS are now highly standardized (e.g., transaction methods, item display, and in-store movement). After entering the store, most consumers can quickly find the goods that they wish to purchase and then directly check out. There are few problems that need to be solved. This is compatible with the characteristics of Taiwanese CVS.

## 6. Conclusions

Most extant research on CVS service quality has used data either from the consumer or the expert perspective. In this study, we collected information from both viewpoints. Applying FSE and AHP, we identified the priorities accorded by different valuable customer groups to CVS service quality indices (quantitative). A higher FSE score indicated a greater effect of CVS service quality on the consumption of a valuable customer group (in Table 5), as well as higher levels of customer satisfaction and loyalty. As such, CVS can discern the needs of service quality of different valuable customer groups. Moreover, CVS can refer to the more detailed weights of different customer groups in Table 3 and allocate limited resources focused on important indices to improve service quality. The end result should be enhanced competitive advantage via augmented efficiency.

This study focused on CVS. Because the market positioning of each type of retailer is different, the weighted results obtained from integrating the opinions of experts and consumers of this industry can only serve as a reference for CVS in Taiwan. Because the groups in the study were consumers, the attributes of the different groups were very similar. Therefore, the values of the final scores of the FSE were close. However, these consumer groups were still able to be distinguished based on the slight differences in the final scores of the FSE.

However, our investigation of service quality in CVS has ample room for further development. Only two layers of factors were structured in the factor table in our work; they addressed only three or four attributes in each criterion. Therefore, some attributes of service quality were likely omitted. Scholars could also collect additional factors and construct an enhanced factor table, as well as categorize customers using other attributes. Future research could also examine different types of businesses than those considered here, as well as undertake empiricism in various countries.

**Author Contributions:** All authors contributed to this manuscript. Conceptualization, K.-Y.H. and H.-P.F.; methodology, K.-Y.H., T.-H.C. and H.-P.F.; writing—original draft preparation, Y.-H.T. and Y.-J.L.; writing—review and editing, T.-H.C. and H.-P.F.; investigation, Y.-H.T. and Y.-J.L.; data curation, Y.-H.T.; visualization, Y.-J.L. All authors have read and agreed to the published version of the manuscript.

**Funding:** This research received no external funding.

**Institutional Review Board Statement:** Not applicable.

**Informed Consent Statement:** Not applicable.

**Data Availability Statement:** Data sharing is not applicable.

**Conflicts of Interest:** The authors declare no conflict of interest.

## Appendix A

The AHP approach applied in this study and the methodology is described in the following seven steps:

(1). Describe of the evaluation issues.

The research issues are the key points of the research discussion and the objective of final evaluation. Therefore, the research issues should be defined specifically to avoid deviation from them as much as possible.

(2). Identify all criteria that affect the issues.

The related performance criteria should be discussed and selected based on the process of reviewing the relevant literature and interviewing experts. These criteria should also be separated in accordance with the level of internal relevance and individual independence.

(3). Construct the hierarchy structure.

A hierarchy structure, in general, can be established from the top through the intermediate levels to the lowest level, which usually contains the list of alternatives. To reduce the complexity of the consistency, the criteria for each alternative should contain no more than seven elements and retain independence individually.

(4). Establish the paired matrices for comparison.

The criteria within each hierarchy should be evaluated against their corresponding criteria in the level above and then compared in pairs between themselves. If there are "*n*" criteria in one hierarchy, decision makers must conduct paired comparisons by *n* (*n*-1)/2. The establishment of paired matrices A leads to determining the weights of the criteria within each hierarchy.

(5). Calculate eigenvectors.

The establishment of paired matrixes is used to obtain the maximum eigenvalues, which should correspond with eigenvectors.

(6). Consistency test.

The purpose of consistency tests is to ensure whether the calculations fit the condition of transitivity in priority. A consistency ratio (*CR*) is used to verify the credibility and reasonability of evaluation, as well as to check whether there is inconsistent causality or conflicts in subjective judgments. The *CR* is acceptable if it does not exceed 0.1 (Saaty, 1980). The definition of the consistency index is as follows:

$$CI = (\lambda_{max} - n)/(n - 1) \text{ and } CR = (CI/RI_n)$$

The positive reciprocal matrix generated by valuation yields different consistency index (*CI*) values at each level. These *R.I.* values are called random indexes. The $\lambda_{max}$ is the maximized eigenvector of a pair-wise comparison matrix. The *n* is an attribute of the matrix, and $RI_n$ is a random index, as shown in Table 1 (Saaty, 1980).

**Table A1.** Random index.

| *n* | 2 | 3 | 4 | 5 | 6 | 7 | 8 | 9 | 10 | 11 | 12 | 13 | 14 | 15 |
|---|---|---|---|---|---|---|---|---|---|---|---|---|---|---|
| *R.I.* | 0 | 0.58 | 0.90 | 1.12 | 1.25 | 1.32 | 1.41 | 1.45 | 1.49 | 1.54 | 1.48 | 1.56 | 1.57 | 1.59 |

(7). Normalization.

This study normalized the weight of the interval level and connected the local weight to acquire the global weights of the criteria in each hierarchy after calculating the weights of all criteria.

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
