# Peer review of "Integrating FSE and AHP to Identify Valuable Customer Needs by Service Quality Analysis"

_sustainability, doi:10.3390/su14031833_

Round 1

Reviewer 1 Report

Report on “Integrating FSE and AHP to Identify Valuable Customer Needs by Service Quality Analysis”

General Comments:

The manuscript presents an application of fuzzy synthetic evaluation (FSE) with Analytic Hierarchy Process (AHP, (Saaty, 2008)) for the understanding of the needs of different valuable customer groups for service quality. Qualitative expert judgements are represented in terms of triangular fuzzy numbers (TFN), i.e. triples of conventional (or: crisp) numbers. Derived weights of the pairwise comparison judgements are employed for ranking priorities for the convenience stores service indices. AHP has been applied to various cases of multi-criteria optimization problems, such as customer satisfaction, development of new products. In this respect, I find interesting the attempt by the authors to use AHP in combination with fuzzy numbers for modeling the decisional process in a convenience stores servicer. In general, this paper can be considered for publication. Formally, the manuscript is quite organized, making it easy to verify. Concerning the contents, I find the methodology also written well so that the main conclusions can be supported. However, I think that in order the manuscript to match the requirements for publication in Sustainability journal, the authors should address the issues listed in the following.

Specific Comments

On page 2, lines 76-77, the authors state that “FSE is suitable for studies that identify valuable customer groups by service quality based on consumer and expert views.” This is the core of the whole manuscript. Although FSE has been discussed in the literature review, to consolidate this argument, the authors should explain why FSE is suitable for this study.

On page 3, lines 102-112, it is suggested to strengthen the description of how to find a valuable customer with RSQS (service quality)

On page 3, lines 125-126, the author should clearly explain the weaknesses of the two methods and how FSE can solve these shortcomings

On page 10, line 343, it is good that readers can clearly understand the differences in customer views through table 7. However, if we want to know the opinions of experts, we must go back to table 5 of page 8, line 283. Can the author also simplify the views of experts in Table 5 into a single index and put it in Table 7 for readers' reference.

On page 11, lines 375-384, It is suggested to explain the way to find out the valuable customers based on table 6 first, and then use table 3 to identify the needs of each group of customers.

Author Response

Dear Reviewer 1

Attached please find the letter of response to your comments.

Hsin-Pin Fu

Reviewer 2 Report

The article is about an interesting subject of the needs of different valuable customer groups for service quality. The paper's idea is clear, but the implementation of the research idea has to be improved.

  1. Introduction
  •  The sentence can be improved: "To survive in such a context, 35 retailers must keep providing both products and services of satisfactory quality." Services are part of the product. Thus it is better to write "goods and services". (Line 36)
  • What is GA? Abbreviations used in the text should be explained. (Line 62)
  • The authors should define more clearly the main aim of the article.
  1. Literature review
  • The title of the paragraph "2.1 Service Quality" doesn't correspond with the content of this paragraph in which authors analyze the measure of the service quality but not service quality itself. (Lines 91 – 112).
  • There is no information about the dynamics of indices. Does it not have any importance? (Lines 166-179).
  1. Methodology
  • The formulas should be written more accurately.
  1. Application
  • The sentence "A total of 304 valid consumer questionnaires were obtained through analysis and 242 manual screening, of which 53 were valid." is unclear. How many respondents should give answers to get reliable results - are 53 respondents enough? Is the AHP method applied correctly? (Lines 242-245).
  1. Result Analysis
  • Table 6 does not show any significant differences in priorities in different customer groups. What is the practical implication of this research? (Lines 363-367).
  • Is the statement that "consumers do not care much about "problem solving"" correct? (Line 368)

Author Response

Dear Reviewer 2

Attached please find the letter of response to your comments.

Hsin-Pin Fu

Reviewer 3 Report

The paper concerns the issue of the Integrating FSE and AHP to Identify Valuable Customer Needs by Service Quality Analysis. The paper proposed a hybrid multiple criteria decision making (MCDM) tool, which uses fuzzy synthetic evaluation (FSE) in combination with the Analytic Hierarchy Process (AHP), to help companies better understand the quantitative data (the weights of the factors that affect service quality) and the qualitative information to identify valuable customers. The following remarks should be referred to: My main concern on the research is related to the limited innovation. What is the value added with respect to existing methods? The relevance of the proposed methodology should be definitely discussed further. Concerning the methodology, although the structure of the AHP is rather clear, there are not enough information on the process to develop it. Particularly, since the authors state that the use of consumers knowledge is a relevant point of their work, I believe that further details are needed on this. More specifically, which consumers were involved? Did you find any differences and/or inconsistencies in their belief?

Author Response

Dear Reviewer 3:

Attached please find the letter of response to your comments.

Hsin-Pin Fu

Round 2

Reviewer 3 Report

The paper concerns the important issue of the Integrating FSE and AHP to Identify Valuable Customer Needs by Service Quality Analysis. The paper proposed a hybrid multiple criteria decision making (MCDM) tool, which uses fuzzy synthetic evaluation (FSE) in combination with the Analytic Hierarchy Process (AHP), to help companies better understand the quantitative data (the weights of the factors that affect service quality) and the qualitative information to identify valuable customers. The choice of reference should be supplemented with respect to the risk analysis methods in water distribution networks, which have been developed (Ref.) Line 162:  Pietrucha-Urbanik, K.; Rak, J.R. Consumers’ Perceptions of the Supply of Tap Water in Crisis Situations. Energies 2020, 13, 3617. https://doi.org/10.3390/en13143617. More detailed future research directions should be added. Besides, there is no discussion about possible limitations of using the proposed approach.

Author Response

The  manuscript has been minor revised
